# A Comparative Analysis of the Wound Healing-Related Heterogeneity of Adipose-Derived Stem Cells Donors

**DOI:** 10.3390/pharmaceutics14102126

**Published:** 2022-10-06

**Authors:** Guoqiang Ren, Qiuyue Peng, Jeppe Emmersen, Vladimir Zachar, Trine Fink, Simone R. Porsborg

**Affiliations:** Regenerative Medicine Group, Department of Health Science and Technology, Aalborg University, Fredrik Bajers Vej 3B, 9220 Aalborg, Denmark

**Keywords:** adipose-derived stem cells, heterogeneity, transcriptome, wound healing, angiogenesis, extracellular matrix

## Abstract

Adipose-derived Stem cells (ASCs) are on the verge of being available for large clinical trials in wound healing. However, for developing advanced therapy medicinal products (ATMPs), potency assays mimicking the mode of action are required to control the product consistency of the cells. Thus, greater effort should go into the design of product assays. Therefore, we analyzed three ASC-based ATMPs from three different donors with respect to their surface markers, tri-lineage differentiation, proliferation, colony-forming unit capacity, and effect on fibroblast proliferation and migration, endothelial proliferation, migration, and angiogenesis. Furthermore, the transcriptome of all three cell products was analyzed through RNA-sequencing. Even though all products met the criteria by the International Society for Cell and Gene Therapy and the International Federation for Adipose Therapeutics and Science, we found one product to be consistently superior to others when exploring their potency in the wound healing specific assays. Our results indicate that certain regulatory genes associated with extracellular matrix and angiogenesis could be used as markers of a superior ASC donor from which to use ASCs to treat chronic wounds. Having a panel of assays capable of predicting the potency of the product would ensure the patient receives the most potent product for a specific indication, which is paramount for successful patient treatment and acceptance from the healthcare system.

## 1. Introduction

Chronic wounds affect millions of people worldwide; they pose a significant burden to the patients and society, and they do not follow the normal wound healing process, resulting in lengthy treatment processes, high treatment costs, and inadequate response to treatment [1,2]. The commonly used clinical treatment for chronic wounds includes inflammation control, wound dressing changing, operative debridement, and flap repair. A new and exciting alternative to the current treatment options is stem cell-based therapy, which stimulates the body’s intrinsic repair mechanism to regenerate tissues and restore normal function. Human adipose-derived stem cells (ASCs), found in adipose tissue, have been acknowledged as one of the sources of clinically relevant stem cells with broad application prospects [3]. The advantages of ASCs are that they are relatively easy to obtain in large numbers from minimally-invasive liposuction procedures, they maintain their characteristics after long-term in vitro.

Culture, and they possess low immunogenicity, which enables the use of allogeneic ASCs [4,5,6]. ASCs have, in preclinical studies, shown great promise as a treatment modality for healing cutaneous wounds [7,8,9], where they have been shown to diminish inflammation [10,11], stimulate angiogenesis [12,13,14,15], and support fibroblast [12,16] and keratinocyte [12,14] growth. It has been hypothesized that ASCs exert their wound healing properties through responsiveness to their environment, leading to stimulation and modulation of the residing tissue cells through the secretion of soluble factors [17].

Large-scale production and subsequent adoption of ASC-based medicinal products require attention to better product standardization, and several production steps, from the isolation of starting material to the storage of the product, could influence the characteristics of the final product and the clinical effectiveness [18]. Indeed, production parameters, such as the use of trypsin or hypoxia, have been shown to influence several wound-healing properties, including enhanced immunosuppression [19], secretion of pro-angiogenesis growth factors [17,20,21,22,23,24], increased endothelial cell growth [17] and angiogenesis [21,22,23,24], re-epithelialization [25,26], and extracellular matrix (ECM) production [27]. Another major area giving rise to variation is the starting material. Several studies have identified a difference in ASCs isolated from different donors based on characteristics, such as age [28,29,30,31,32,33], sex [34,35], and body mass index (BMI) [36,37], influencing the proliferation [28,29,31], viability [29], migration [32], angiogenic [33,36], multilineage differentiation [28,29,30,34,37] capacity of the ASCs. Recently, ASCs isolated from aged and diabetic donors showed a reduced capacity for accelerating fibroblast migration and endothelial cell angiogenesis compared to younger donors [32]. Furthermore, it has been suggested that specific genetic profiles of individual donors may introduce a non-negligible inconsistent therapeutic effect of cells when used for clinical purposes [38].

However, no previous study has yet linked these donor characteristics to a molecular biomarker of ASCs performance in relation to specific applications, such as wound healing. A biomarker (short for molecular biomarker) is a biological indicator of a biological state or condition, as defined by World Health Organization (WHO) as “any substance, structure, or process that can be measured in the body or its products and influence or predict the incidence of outcome or disease” [39]. Such a marker could be used for stem cell production to ensure the selection of the donors giving rise to the most potent ASCs and thereby optimizing the chance for patients to receive the best treatment modality [18].

Therefore, in this study, we aim to identify a panel of markers to use in the future screening of ASCs for downstream production. We comprehensively characterize ASCs and their inter-donor variations based on a panel of quantitative characteristic and functional parameters, using well-established in vitro models [40,41], representing various wound healing processes. We link the identified differences in the effect size of the specific mode of action (MoA) to molecular targets of the mechanism of action (MeA) identified in the transcriptome of the ASCs.

## 2. Materials and Methods

### 2.1. Cell Isolation and Culture

Adipose tissue was anonymously donated by three healthy subjects undergoing cosmetic liposuction surgery at Aleris Private Hospital, Aalborg, Denmark (Gender: male, female, female; Age: 44, 53, 49; Location: abdomen, abdomen, knee; BMI: 25.4, NA, 23.6. All the donors had signed an informed consent form, and the protocol was approved by the regional committee on biomedical research ethics of Northern Jutland, Denmark (Project no. N-20160025).

ASCs were isolated, as previously described [42]. In brief, after being washed four times with sterile phosphate-buffered saline (PBS) (Invitrogen, Taastrup, Denmark), the adipose tissue was dissociated with 0.6 U/ml Collagenase NB 4 Standard Grade (Serva Electrophoresis, Heidelberg, Germany) in Hanks balanced salt solution (HBSS; Invitrogen) for 60 min at 37 °C under continuous agitation. After filtering the dissociated tissue through a 100-μm filter (Millipore, Omaha, NE, USA) and centrifugation at 400× *g* for 10 min, the pellet was resuspended and filtered through a 60-μm filter and pelleted again by centrifugation at 400× *g* for 10 min. The obtained cell pellet was handled and expanded in an automated bioreactor Cell Expansion system (Terumo BCT, Lakewood, CO, USA) and cryopreserved as described by Haack-Sørensen et al. [40]. For the subsequent analysis of the ASCs, the cells were thawed and cultured in T175 culture flasks (Greiner Bio-one, Frickenhausen, Germany) in alpha-Minimum Essential Medium with GlutaMAX supplemented with 10% fetal calf serum (FCS) and 100 U/ml penicillin and 0.1 mg/mL streptomycin (all from Invitrogen), and the TrypLE (Gibco) was used to detach the cells when they reached 70–80% confluency. Hereafter, the ASC cultures were named ASC-101, ASC-105, and ASC-106 and used in passages 1–2 for subsequent procedures. For an overview of the experimental setup, see Figure 1.

Human Dermal Microvascular Epithelial Cells (HDMECs, Promo-cell, Heidelberg, Germany) and Human Dermal Fibroblasts (HDFs, Life Technologies, Frederick, MD, USA) were cultured following the manufacturer’s instructions.

### 2.2. Stem Cell Characterization

To characterize the surface marker phenotypes of the three ASC populations, multichromatic flow cytometry was used. The detailed protocol has been described previously by our laboratory [43]. In brief, passage 2 ASCs were stained with antibodies against the surface markers CD29, CD90, CD166, CD73, CD201, CD248, CD105, CD274, CD34, CD271, CD146, CD146, Stro-1, CD36, CD200, and CD31 for 30 minutes (BD Biosciences, Lyngby, Denmark). After washing twice with PBS, the fluorescence of the ASCs was measured by the CytoFLEX flow cytometer (Beckman Coulter, Copenhagen, Denmark) and analyzed using the Kaluza 2.1 software package (Beckman Coulter, Copenhagen, Denmark).

### 2.3. Trilineage Differentiation

#### 2.3.1. Adipogenic Differentiation

The three ASC populations were seeded in 12-well tissue culture plates at 5000 cells/cm^2^ and incubated for 2 weeks with the STEMPRO^®^ Adipogenesis Differentiation Kit (ThermoFisher Scientific, Life Technologies, Roskilde, Denmark). Hereafter, the cells were fixed in 4% paraformaldehyde (AppliChem, Esbjerg, Denmark) in PBS, and the lipids were stained using Oil Red O (Sigma-Aldrich, Søborg, Denmark) for 30 min. Images of the tri-lineage differentiation were captured by standard bright field microscopy (Olympus CKX41, Life Science Solution, Ballerup, Denmark).

#### 2.3.2. Osteogenic Differentiation

ASCs were seeded in 12-well tissue culture plates at 2500 cells/cm^2^ and incubated for three weeks with the STEMPRO^®^ Osteogenesis Differentiation Kit (ThermoFisher Scientific, Life Technologies, Roskilde, Denmark). The cells were then fixed in 4% paraformaldehyde, and the calcium deposits were stained with Alizarin Red (Sigma-Aldrich) for 30 min.

#### 2.3.3. Chondrogenic Differentiation

ASCs were seeded in a 96-well V-shaped tissue culture plate (Greiner Bio-one, Frickenhausen, Germany) at 80,000 cells per well, centrifuged at 500× *g* for 5 min, and incubated for 3 weeks with the STEMPRO^®^ Chondrogenic-differentiation kit (ThermoFisher Scientific, Life Technologies, Roskilde, Denmark). The control pellets were maintained in a growth medium. Hereafter, the pellets were embedded in 5 µm paraffin sections, the sulfated glycosaminoglycans (GAGs) stained with Alcian Blue 8GX (Sigma-Aldrich, Søborg, Denmark) for 30 min, and the degree of differentiation was evaluated by bright field microscopy (Axio Observer).

### 2.4. Colony-Forming Unit

To evaluate the frequency of colony-forming unit (CFU) in the three ASC cultures, each culture was seeded in a limiting dilution assay at densities of 1–30 cells per well in a 96-well plate (Costar, Corning Life Sciences, Tewksbury, MA, USA) and cultured for 14 days with medium change twice a week. On day 14, the cells were fixed in 4% formaldehyde and stained with 0.05% Crystal Violet (Sigma Aldrich). Wells containing one or more colonies were counted, and the proportion of CFUs was assessed based on the Poisson statistics using L-Calc software (Stem Cell Technologies, Vancouver, BC, Canada).

### 2.5. Proliferation

To quantify the proliferation of ASCs, these were seeded with 600 cells/cm^2^ in a 96-well plate to enable the coverage of the exponential phase of the proliferating cell cultures and cultured in ASC culture medium for 13 days. To measure the effect of ASC-CM on the proliferation of HDFs and HDMECs, these were seeded in 96-well plates at a density of 600 and 15,000 cells/cm^2^, respectively, in either ASC-CMs or a non-conditioned ASC medium (MEM) as vehicle control. The cells were hereafter cultured for 13 or 5 days, respectively, with media change every 3 days. For each cell type, cells were lysed on days 1, 3, 5, 7, 9, 11, and 13 or on days 1, 3, and 5 using 0.02% SDS (Sigma-Aldrich) for 30–40 min and frozen. The DNA content was quantified using a QUAN-IT PicoGreen kit (ThermoFisher Scientific) and a lambda DNA standard curve according to the manufacturer’s protocol. The fluorescence was examined using an EnSpire multimode plate reader (PerkinElmer, Boston, MA, USA) at 485/535 nm.

### 2.6. Migration

Monolayers of confluent HDFs and HDMECs in 96 wells were scratched using the Wound Pin tool (V&P Scientific, Radway Green, UK) and washed with PBS to remove cell debris. The ASC-CMs were added as aforementioned, and the MEM culture medium was used as vehicle control. An inverted light microscope with a phase-contrast camera (Olympus CKX41, Life Science Solution, Ballerup, Denmark) was used to take pictures at 0, 6, and 12 h for HDFs and 0, 24, and 48 h for HDMECs, and cell migration was analyzed using the Image J software (https://imagej.nih.gov/ij/ (accessed on 16 June 2022)).

### 2.7. Tube Formation

To model 3D angiogenesis, HDMECs were seeded at 10,000 cells per well in a 48-well plate coated with a 130 µL Basement Membrane Matrix (BD Biosciences, CA, USA). ASC-CM from each donor was added, and MEM was used as vehicle control. Images were captured after 4 h of incubation using a brightfield microscope and analyzed using the Image J software plugin-Angiogenesis analyzer.

### 2.8. RNA Sequencing

Total RNA was isolated using the Aurum Total RNA Mini Kit (Bio-Rad, Copenhagen, Denmark) according to the manufacturer’s instructions, and RNA sequencing (RNAseq) was performed by the BGI. Europe A/S (Copenhagen, Denmark). The mRNA enrichment and purification using oligo (dT) magnetic beads and cDNA synthesis were operated, followed by PCR amplification. Hereafter, the final library products were sequenced using BGIseq-500. The expression levels for each gene were normalized to fragments per kilobase of exon model per million mapped reads (FPKM) using RNA-seq by Expectation-Maximization (RSEM).

The GO bioinformatics and heat map gene expression analyses were carried out using the visualization tool Dr. TOM (BGI). The upregulated or downregulated expression of genes was shown as log2FC, which represents log-transformed fold change (log2FC = log2[B] − log2[A]), while A and B represent values of gene expression for different treatment conditions. Hereafter, the molecular targets identified from the RNAseq were thoroughly reviewed by a literature search to find existing evidence of their MeA in relation to stem cell function and wound healing, and, based upon this, narrowed down to a suggestive panel for further analysis.

### 2.9. Statistical Analysis

All data are presented as mean ± SEM. The student’s t-test analyzed a comparison between two groups. When comparing more than two groups, one-way or two-way ANOVA with a Bonferroni post-hoc test was used. A *p* value < 0.05 was considered statistically significant. Statistical analysis was conducted by GraphPad Prism 9 software and SPSS version 27.0.

## 3. Results

### 3.1. Identification of ASCs and Characteristic Comparison

ASCs cultures from three distinct donors were isolated, and a trilineage differentiation assay was carried out to confirm their multipotency. Histochemical analysis showed that all cultures could differentiate into both adipocytes, osteocytes, and chondrocytes (Figure 2A). The colony-forming potential was found to be higher for ASC-106 when compared to ASC-105 and ASC-101 (0.952 ± 0.027 vs. 0.363 ± 0.034 and 0.227 ± 0.02, respectively; both *p* < 0.0001) (Figure 2B). Furthermore, the colony-forming potential of ASC-105 was found to be higher than that of ASC-101 (0.363 ± 0.034 vs. 0.227 ± 0.02; *p* = 0.012). The proliferative capacity of ASC-106 and ASC-105 was comparable and both prominently faster than that of ASC-101, giving rise to significantly more cells at day 13 (477.43 ± 19.52 and 376.63 ± 11.93 vs. 302.23 ± 12.63, respectively; *p* = 0.0004 and 0.03, respectively) (Figure 2C). To evaluate the immunophenotypical profile of the ASCs, a multichromatic flow cytometry assay was performed (Figure 2D). The surface markers evaluated were categorized into five groups, based on their functionality: Mesenchymal stem cell (MSC) markers, ASC markers, differentiation capacity, immune regulation, and wound healing (Table 1). For a more detailed description of the rationale behind the selection of these markers, please refer to our previous paper [43]. The results showed that all ASC cultures were strongly positive for CD90, CD105, and CD73, recognized as the quintessential MSC markers. For the ASC marker group, all ASC cultures were positive for CD34, weakly positive for CD146, and negative for CD31, especially in ASC-101 (0.77%). In the markers relevant for ASC differentiation capacity, all ASC cultures were strongly positive for CD201 and weakly positive for CD36 and Stro-1. ASC markers connected to immune regulation showed CD29 and CD274 to be strongly positive in all ASC cultures, but CD200 was only weakly positive. In the markers related to wound healing, all ASC donors were strongly positive for CD166 and CD248 and positive for CD271.

### 3.2. Donor-Dependent Difference in ASC Function during Wound Healing

To explore the difference in performance between ASC cultures derived from the different donors, ASC-CM was produced from each culture and compared in a collection of in vitro wound healing assays based on fibroblasts and endothelial cells, mimicking suggested modes of action. Biological assays relevant to the granulation phase of wound healing were fibroblast migration and proliferation assays (Figure 3). For measuring the migration rate of fibroblasts, these were cultured to form a confluent monolayer, which was subsequently scratched and supplied with either MEM, as vehicle control, or conditioned media derived from ASC-101, -105, and -106. Hereafter, the cell-free area was measured every 6 hours until total closure was observed. For representative images of the scratch closure, see Figure 3A. The quantitative evaluation of the scratch closure revealed CM derived from ASC-106 and -105 to increase the migration of fibroblasts to a higher degree than that derived from ASC-101, resulting in significantly faster closure of the scratch (*p* < 0.001 and = 0.03, respectively) (Figure 3B). For evaluating the effect of ASC-CM on the proliferation of fibroblasts, they were cultured for 13 days in MEM or CM derived from ASC-101, -105, and -106, and the total DNA content evaluated every second day as a measure of total cell number. From this, it was evident that the cell number of fibroblasts cultured in CM derived from ASC-106 was significantly higher than that of fibroblasts cultured in CM from ASC-105 or -101 (*p* < 0.001 and =0.02, respectively) (Figure 3C). Furthermore, CM derived from ASC-105 was found more stimulatory than that from ASC-101 (*p* = 0.002).

Assays relevant for the process of angiogenesis included endothelial proliferation, migration, and tube formation assays. When analyzing the effect of ASC-CM on the migration of endothelial cells through a scratch assay, CM derived from ASC-106 stimulated the scratch closure significantly more than that derived from ASC-105 and -101 (both *p* < 0.001) (Figure 4A,B). In accordance with this, CM from ASC-106 significantly outperformed that of ASC-105 and -101 when measuring endothelial cell proliferation (both *p* < 0.001) (Figure 4C). As a model of 3D angiogenesis, a tube formation assay was employed, and the complexity of tube networks was evaluated after 4 hours (Figure 4D). Here, it was found that CM derived from ASC-106 and ASC-105 was superior to that of ASC-101 in all measured parameters, including branching length, extreme number, node number, and junction number (Figure 4E).

### 3.3. Differences in Gene Expression between Donors and GO Analysis

Based on the characteristic and functional tests of the ASC cultures derived from distinct donors, it was found that ASC-105 and –106 might present beneficial properties in comparison to ASC-101, which might be explained by their proliferative capacity or other functional attributes. To further understand the underlying mechanism of action behind this, samples from each culture were analyzed by total RNA sequencing to explore the transcriptomic differences between these. For all three cultures, 230,487 mRNA transcripts (24,237 genes) were identified and the expression level quantified. When analyzing significantly upregulated genes (log2FC ≥ 1, Q-value ≤ 0.05) of ASC-106 and ASC-105 compared to the expression level of ASC-101, 158 genes were simultaneously upregulated in both ASC-106 and ASC-105 (Figure 5A).

Next, gene ontology (GO) analysis of the biological processes associated with all the upregulated genes, demonstrated upregulation of 31 genes in categories relevant to wound healing, such as ECM development, angiogenesis, the proliferation and migration of cells, and stem cell development, such as cell differentiation, and the rest of the genes clustered in other categories (Figure 5B).

A separate analysis was based only on the most significantly upregulated genes, which comprised a set of 14 genes (Figure 5C). This prominent cluster of 14 genes was analyzed by GO Biological Process ontology (Figure 5D), which demonstrated that most of the 14 genes were relevant to wound healing. Jointly, these two approaches demonstrated that GO terms pertinent to wound healing, such as ECM development, angiogenesis, the proliferation and migration of cells, and stem cell development, such as cell differentiation, were among the most significantly regulated categories in ASC-106 and ASC-105 compared to ASC-101.

When analyzing the genes with a significantly higher expression in ASC-101 (log2FC ≥ 1, Q-value ≤ 0.05) compared to ASC-106 and ASC-105, several genes were found linked to processes involved in pre-adipocyte-formation or adipogenesis: *CD24, CDSN, OMD, ACTC1, COL4A5*, and *LAMB3*. This could indicate that ASC-101 was more committed toward adipogenic differentiation and thereby not as naïve as ASC-105 and ASC-106. Furthermore, the expression of *CDH6, COL6A6, ANOS1, TINAGL1, DGCR6, CD4, ADGR1, CLCA2, ACTG2, FOXF1, ECM2, ACAN, PECAM1, TNXB,* and *COL14A1* linked to processes involved in cell adhesion, ECM organization, and mesenchyme migration, were higher in ASC-101 (data not shown).

### 3.4. Selection of Putative Markers Predicting Wound Healing Capacity

To select the most promising markers predicting wound healing capacity, putative predictors were selected based on the analyses above (Figure 6). In summary, from the pool of 230,487 mRNA transcripts (24,237 genes), 158 genes (Appendix A) upregulated in both ASC-105 and -106 compared to ASC-101 were identified (Figure 5A). Next, a combination of the two different analytical approaches were used to select genes of interest among the upregulated genes. From the first approach based on the gene ontology (GO) analysis of all genes (Figure 5B), 31 genes associated with wound-healing-related processes were selected. From the second approach based on the heat map analysis, we identified the 14 most significantly upregulated genes (Figure 5C). Interestingly, between these two approaches, there was an overlap of 10 genes, leading to a total of 35 unique genes (Table 2). Based on a thorough literature review of the 35 genes, we further narrowed the list down to 10 genes of special interest comprising transcription factors (TBX1, EPAS1), membrane proteins (FGFR2, ITGB8, GREM1), and secreted proteins (MMP1, MMP9, COL4A4, FGF9, and CCL11).

## 4. Discussion

In recent years, stem cell-based therapies are slowly gaining ground in routine medical care due to their capability to repair or replace damaged tissue, as a result of their natural ability to produce cytokines and other molecules, especially in skin wound management. ASCs, as one of the clinically most promising stem cell types, have been put in the spotlight in a broad range of clinical studies as a treatment modality for healing cutaneous wounds (clinicaltrials.gov, NCT01932021, NCT02099500, NCT02092870, NCT02314416, NCT02394873). A series of preclinical studies have shown variability between ASCs derived from different donors in terms of proliferation and trilineage differentiation capacity [29,30,31,34,41]. Thus, to strengthen the success rate of implantation and treatment effect for later clinical use, the therapeutic potential of ASCs in terms of wound healing and their inter-donor variations need to be explored in more detail. In this study, we investigated the heterogeneity of ASC from different donors based on a set of characteristic and functional assays related to stem cell characteristics and their mode of action within wound healing. Additionally, to get closer to the mechanism of action, we examined the regulation of genes involved in these processes. Our results indicate that specific regulatory genes associated with ECM and angiogenesis could be used as markers of a superior ASC donor from which to use ASCs to treat chronic wounds.

When characterizing the ASCs from the three distinct donors, our approach was based on the criteria suggested in the position statement from the International Federation for Adipose Therapeutics and Science (IFATS) and the International Society for Cellular Therapy (ISCT) [42]. The ASCs from all donors had the capability to undergo tri-lineage differentiation, had a high proportion of CFUs and were capable of proliferating. Furthermore, all three cultures were positive for the characteristic markers CD105, CD73, CD90, and our selected ASC-markers CD146 and CD34. Moreover, a very low expression of CD31 was seen. However, it was evident that both ASC-105 and -106 had a higher proliferation rate and proportion of CFU when compared to ASC-101. It could therefore be suspected that ASC-105 and -106 could have a higher potency than ASC-101. When analyzing the transcriptome of the three ASC cultures, several transcripts giving rise to membrane-bound proteins were identified. The transcripts of fibroblast growth factor receptor 2 (*FGFR2*), integrin subunit beta 8 (*ITGB8*), and protein tyrosine phosphatase receptor type B (*PTPRB*) were enriched in both ASC-105 and -106 when compared to ASC-101. FGFR2 is a member of the fibroblast growth factor receptor family and generally plays a vital role in stimulating angiogenesis by binding with FGF2, promoting keratinocytes proliferation and the regulation of osteoblast differentiation, proliferation, and apoptosis [43,44,45]. In ASCs, FGF2 has been linked to promoting the proliferation ability of hASCs via Src activation [46]. ITGB8 is a member of the integrin beta chain family and plays a role in cell migration and cell–cell and cell–ECM adhesion [47,48,49]. In addition, ITGB8 is a receptor for fibronectin, which plays a significant role In cell growth, migration, adhesion, and differentiation, and facilitates wound healing [50]. Specifically for mesenchymal stem cells, ITGB8 has been shown to promote chondrogenic differentiation [51] and has been found differentially expressed in ASCs from different donors [32]. PTPRB, a member of the protein tyrosine phosphatase (PTP) family, is an enzyme mainly expressed in endothelial cells essential for blood vessel remodeling and angiogenesis [49]. In ASCs, it has been linked to reduced adipocyte differentiation [52]. Moreover, the transcripts T-box transcription factor 1 *(TBX1)* and endothelial PAS domain-containing protein 1 *(EPAS1)* were identified, both giving rise to transcription factors. TBX1 is essential in forming tissues and organs during embryonic development [53]. Likewise, TBX1 has been identified as a novel regulator of energy homeostasis, metabolic signaling, and adipocyte growth, development, and differentiation. However, exactly how TBX1 regulates metabolic signaling in subcutaneous adipose tissue is still not fully elucidated. It could be through direct DNA binding and/or through off-DNA protein-protein interactions [54]. EPAS1, also well-known as hypoxia-inducible factor-2alpha (HIF-2α), is involved in the induction of oxygen-regulated genes. It has been noted to contribute to regulating stem cell function and differentiation through activation of Oct-4 [55]. Besides, EPAS1 (HIF-2α) has been verified to facilitate the preservation of stemness and to promote stem cell proliferation of human placenta-derived mesenchymal stem cells [56].

When analyzing the functionality of the ASCs, assays relevant for both the granulation tissue and angiogenesis were included. For the formation of granulation tissue, we evaluated the inter-donor variability using fibroblast proliferation and migration assays. Fibroblasts play a vital role in the proliferation phase of wound healing by proliferating and migrating to the wounded area. We found that conditioned mediums from ASC-106 and 105 were superior to ASC-101 for both accelerating fibroblast proliferation and migration. To identify the molecules involved in this variability, RNA sequencing and subsequent GO analysis of the biological process common for the co-upregulated genes revealed *FGF9, GREM1,* and *CCL11* to be enriched in relation to the wound healing GO term. FGF9 has been shown to increase fibroblast migratory capacities [57]. In addition, *GREM1* encoding gremlin-1 is a member of the BMP (bone morphogenic protein) antagonist family. Gremlin-1 has been demonstrated to promote the proliferation of intestinal fibroblasts [58]. Likewise, overexpression of Gremlin-1 has been shown to entail TGF-β pathway elevated ECM production and myofibroblast transition [59,60,61]. CCL11, encoding C-C motif chemokine 11, also known as eotaxin-1, facilitate the migration and proliferation of lung fibroblasts [62]. Furthermore, fibroblasts synthesize and modulate the ECM components. These create a scaffolding-like structure, providing the structural integrity for the formation of granulation tissue and supporting adhesion and migration of cells as macrophages, endothelial cells, and fibroblasts [63,64]. In this study, RNA sequencing and subsequent GO analysis of the biological process typical for the co-upregulated genes also revealed *MMP1* and *MMP9* to be partially co-upregulated. It has formerly been found that MMP9 stimulates dermal fibroblast migration [65] and contributes to synovial fibroblast proliferation and migration [66]. MMP9 also facilitates cell migration by degrading collagen type IV, thereby breaking down the basement membrane [67]. MMP1 usually plays a role in wound healing by promoting keratinocyte migration on fibrillar collagen, but it has been found to contribute to lung fibroblast migration in epithelial cells [68,69]. However, these genes have also been linked to fibrosis and dysregulation of the wound healing response, emphasizing the importance of a controlled regulation of these [70,71,72].

For angiogenesis, we found evidence of both direct and indirect mechanisms of action. As a direct measure of angiogenesis, the effect of ASCs from different donors on endothelial cell proliferation, migration, and tube formation was used. Overall, our data showed that CM from ASC-106 and 105 possessed a superior ability compared to CM from ASC-101 to promote both endothelial cell proliferation, migration, and angiogenesis. Previous work showed that ASCs from leaner-weight donors showed a higher angiogenic potential than heavier ones [36]. When diving into the underlying mechanism of action behind these effects, RNA sequencing and subsequent GO analysis of the biological process for the co-upregulated genes found evidence supporting this phenomenon. The gene expression of the superior ASCs, ASC-106 and 105, were enriched within processes of angiogenesis, endothelial cell proliferation, and endothelial cell migration regulation. Among the enriched genes was *FGF9* and the lack of FGF9 in vivo has been verified to decrease angiogenesis [73]. In general, FGF family members have shown an essential role in wound healing, as evidenced by stimulating angiogenesis [73,74,75,76,77,78]. Furthermore, a significant increase was found in Gremlin 1 (*GRME1*), which is involved in angiogenesis-modulating in endothelial cells, endothelial cell migration, and cell migration involved in sprouting angiogenesis [79,80]. Another interesting gene upregulated is *CCL11*, a direct mediator of angiogenesis, as evidenced by its ability to induce in vitro endothelial cell proliferation, migration, and angiogenesis [81,82]. 

An indirect influence of ASCs on the process of angiogenesis could be through the regulation of ECM. The ECM not only provides a scaffold to support the formation of connective tissue but also plays a crucial role in angiogenesis by supporting the migration of endothelial cells to form new blood vessels in the wound bed [83]. In this study, we found a significant upregulation in ASC-105 and -106 of several ECM-relevant genes, including *MMP1*, *MMP9*, and *COL4A4*, which have all been shown to play an important role in angiogenesis. In support of this, studies found that MMP-mediated ECM degradation by ASCs accelerates the angiogenesis of endothelial cells [84,85]. MMP1 has been found to trigger vascular remodeling and angiogenesis in the wound healing process by increasing the expression of the vascular endothelial growth factor receptor 2 (VEGFR2) and endothelial cell proliferation [86]. MMP9 has been validated to increase the secretion of VEGF from endothelial cells, accelerate endothelial cell migration and stimulate vessel growth [87]. *COL4A4* encodes the alpha4 (IV) chain of type IV collagen. Type IV collagen is an essential component of the endothelial basement membrane to support the formation of blood vessels and contributes to endothelial cell migration [88,89]. Additionally, it has been proven to facilitate the new microvascular elongation and survival in a dose-dependence manner in an ex vivo aortic ring model [90]. These upregulated genes could putatively be potential screening markers for ASCs with superior performance in relation to angiogenesis in wound healing.

When analyzing the inter-donor variation of ASCs from the perspective of wound healing, angiogenesis and ECM-relevant processes and genes were the main groups enriched. Based on their biological significance and the level of existing evidence, we propose a panel of stem cell- and wound healing-related genes, especially related to ECM and angiogenesis, to further validate the future screening of ASC-based medicinal products for wound healing. The suggested panel comprises *MMP1*, *MMP9*, *COL4A4*, *ITGB8*, *TBX1*, *EPAS1, FGFR2*, *FGF9*, GREM1, and CCL11. However, it should be mentioned that any of the 35 genes identified could hold the potential as a marker of potency. Furthermore, it should be noted that this study has examined the ASC-donor variability in vitro and by RNA-Seq technology used only to draw attention to the differences in the gene expression in ASCs from different donors, which could be used to explain the observed difference in effect. Therefore, multiple validations and further studies based on in vivo assays are required to shed more light on the biological difference between donors in wound healing and to fully validate these markers to be directly linked to the mechanism of action of ASCs in wound healing. Ultimately, the goal would be to identify markers predictive of the wound healing capacity of a stem cell preparation in a clinical setting, such that those markers could be used as potency assays. The importance of developing validated potency assays is underscored by the fact that both the European Medicines Agency (EMA) and the U.S Food and Drug Administration require such assays as part of the release criteria after transitioning from early phase trials to an approved medicinal product.

## Figures and Tables

**Figure 1 pharmaceutics-14-02126-f001:**
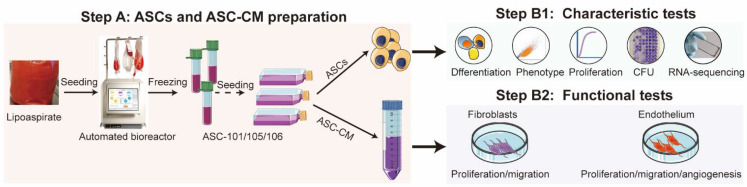
Flowchart of the workflow process. Step A: ASCs from three donors were isolated from lipoaspirate, expanded in an automated bioreactor, and frozen. After thawing, the cells were expanded in flasks from which ASCs were harvested for characteristic tests (Step B1), and conditioned media (ASC-CM) was harvested for functional tests (Step B2). To characterize the ASC, ASCs were seeded at 8,000 cells cells/cm^2^ in separate T175 culture flasks (Greiner Bio-one, Frickenhausen, Germany), cultured until 80% confluent, washed three times with PBS and then supplied with fresh culture media. After 24 h, the CM was collected, centrifugated at 400× *g* for 10 min, and the supernatant frozen at −80 °C.

**Figure 2 pharmaceutics-14-02126-f002:**
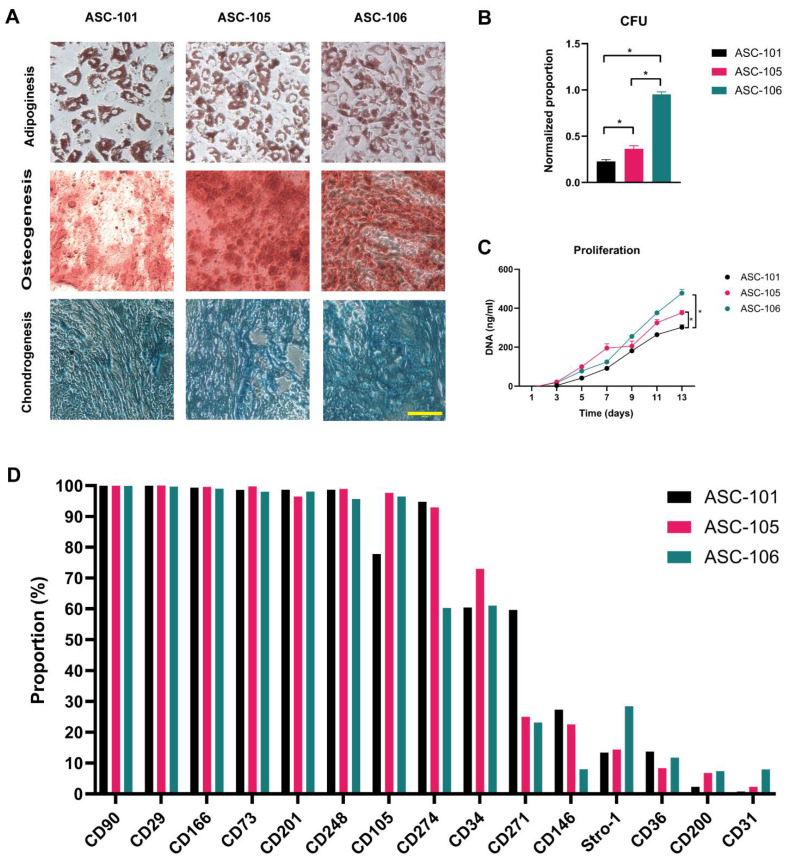
Characteristic tests of adipose-derived stem cells (ASCs); (**A**): Histochemical analysis of Table 1. µm. (**B**): The colony-forming capacity (CFU, *n* = 5). (**C**): The proliferation of ASC cultures (*n* = 3). (**D**): ASCs surface markers evaluated by Flow cytometry. The data are presented as mean + SEM, * indicates a statistically significant change, *p* < 0.05. CFU, colony-forming unit.

**Figure 3 pharmaceutics-14-02126-f003:**
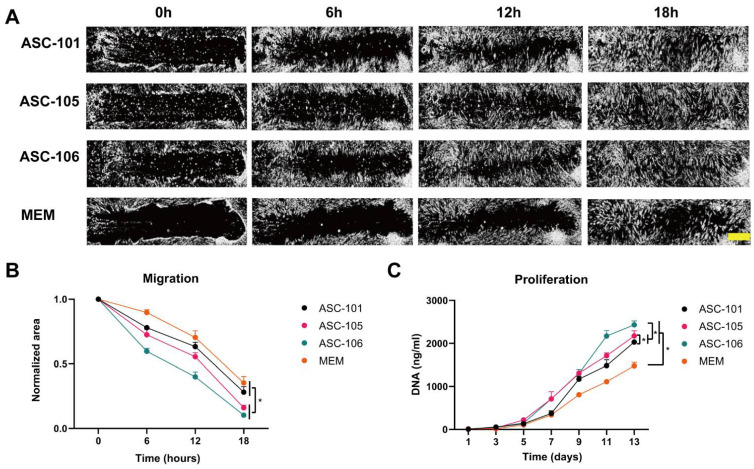
The effects of conditioned media from different ASC cultures (ASC-CM) on the migration and proliferation of dermal fibroblasts. (**A**): Representative images from different time points of the closure of scratched dermal fibroblasts after ASC-CM treatment in relation to the effect of unconditioned media used as vehicle control (MEM). Scalebar denotes 100 µm. (**B**): Quantitative assessment of the effect of ASC-CM on the development of scratch size over time (*n* = 8). The area is normalized to the starting area of each scratch. (**C**): The proliferation of dermal fibroblasts, quantified by the total amount of DNA, after ASC-CM treatment (*n* = 3). The data are presented as mean + SEM, * indicates a statistically significant change. *p* < 0.05. ASC, adipose-derived stem cell; MEM, Minimum Essential Media.

**Figure 4 pharmaceutics-14-02126-f004:**
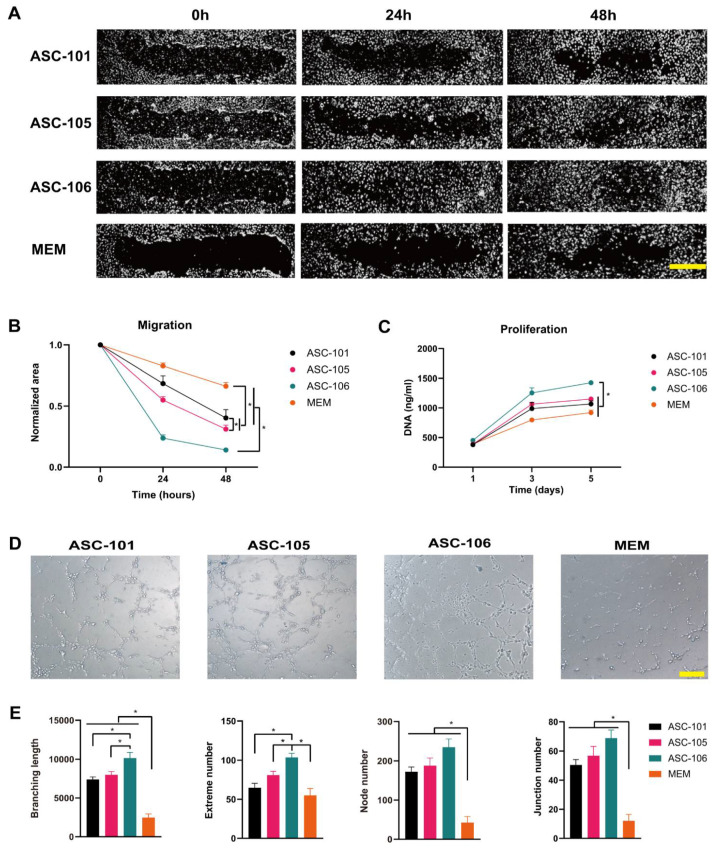
The effects of conditioned media from different ASC cultures (ASC-CM) on migration, proliferation, and angiogenesis of human dermal microvascular endothelial cells (HDMECs). (**A**): Representative images of the closure of scratched HDMECs over time. Scalebar denotes 100 µm. (**B**): Quantification of the effect of ASC-CM on the scratch closure (*n* = 8). The scratch area is normalized to the starting size of the scratch. (**C**): The effect of ASCs on the HDMECs proliferation (*n* = 3). (**D**): Representative images of tube formation of HDMECs after ASC-CM treatment or the vehicle control; a non-conditioned ASC medium (MEM). Scalebar denotes 250 µm. (**E**): Quantification of the effect of ASC-CM on angiogenesis by means of the total branching length, extreme, node, and junction number (*n* = 5). The data are presented as mean + SEM, * indicates a statistically significant change, *p* < 0.05. ASC, adipose-derived stem cell.

**Figure 5 pharmaceutics-14-02126-f005:**
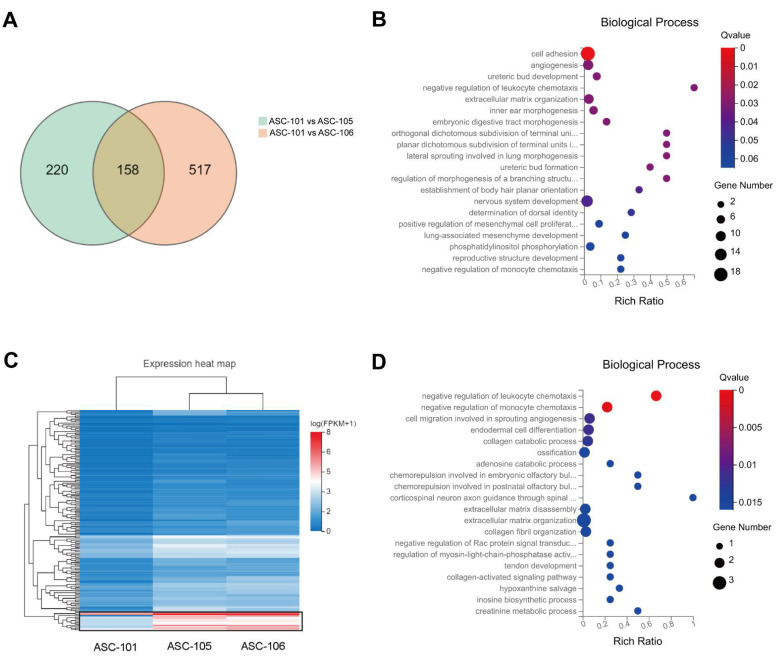
Variability in gene expression profiles between ASC-donors (**A**): A Venn diagram showing the number of upregulated genes and the shared upregulated genes in ASC-105 and -106 compared to ASC-101. (log2FC ≥ 1, Q value ≤ 0.05). (**B**): Gene ontology (GO) analysis of the biological process of the 158 co-upregulated genes. (**C**): The heatmap shows the hierarchical clustering of the 158 genes co-upregulated in ASC-105 and ASC-106 compared to ASC-101. The black box highlights the significantly upregulated genes. (**D**): GO analysis of the biological process of the significantly upregulated genes highlighted the heatmap. ASC, adipose-derived stem cell.

**Figure 6 pharmaceutics-14-02126-f006:**
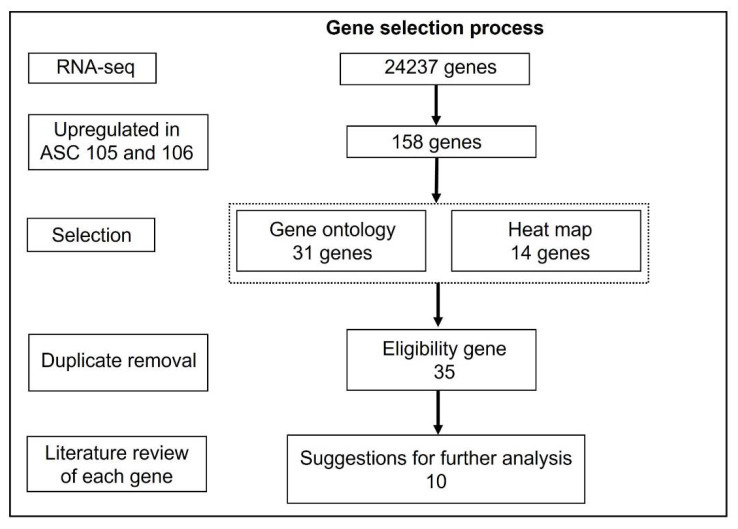
Outline of the selection process of putative markers predictive of wound healing capacity. RNA-seq, Ribonucleic acid sequencing; ASC, adipose derived stem cell.

**Table 1 pharmaceutics-14-02126-t001:** ASCs surface marker expression.

Function	Marker	ASC-101	ASC-105	ASC-106
MSC markers	CD105	77.73	97.65	96.43
	CD73	98.60	99.71	97.99
	CD90	99.95	99.96	99.92
ASC markers	CD146	27.33	22.52	7.99
	CD34	60.41	72.94	61.01
	CD31	0.77	2.33	7.94
Differentiation capacity	Stro-1	13.38	14.38	28.36
	CD201	98.65	96.44	98.08
	CD36	13.67	8.33	11.74
Immune regulation	CD29	99.99	100	99.69
	CD200	2.29	6.77	7.37
	CD274	94.79	92.93	60.28
Wound healing	CD248	98.68	98.87	95.66
	CD166	99.39	99.62	99.03
	CD271	59.61	24.98	23.12

MSC, Mesenchymal stem cell; ASC, adipose-derived stem cell.

**Table 2 pharmaceutics-14-02126-t002:** Enriched biological processes based on gene ontology terms for co-upregulated genes of ASC-105 and ASC-106 compared to ASC-101, with Q-value < 0.05.

Description	Gene	Go Term
**ECM and tissue development**		
Cell adhesion	*TENM2, HAPLN1, SVEP1, PCDHB3, PGM5, CADM3, PCDHB2, SPON1, EFS, CCL11, LZTS1, ITGB8, CELSR1, NUAK1, PDZD2, ITGA11, PCDH19*	GO:0007155
Cell-matrix adhesion	*ITGB8, ITGA11*	GO:0007160
ECM	*MMP1, LOC102724770-DGCR6, COL4A4, HAPLN1, RARRES2,*	GO:0031012
ECM disassembly	*MMP1*	GO:0022617
ECM organization	*MMP9, TNFRSF11B, COL10A1, COL11A1, ITGB8, ITGA11*	GO:0030198
Extracellular region	*ADA2*	GO:0005576
Collagen-containing extracellular matrix	*MMP9, COL10A1*	GO:0062023
Collagen fibril organization	*COL11A1*	GO:0030199
Collagen catabolic process	*MMP1*	GO:0030574
**Angiogenesis development**		
Angiogenesis	*FGF9, TBX1, FGFR4, GREM1, FGFR2, EPAS1, PTPRB*	GO:0001525
Positive regulation of angiogenesis	*PIK3R6, GREM1, CCL11, ITGB8*	GO:0045766
Endothelial cell morphogenesis	*COL4A4*	GO:0001886
Blood vessel morphogenesis/development	*TBX1*	GO:0048514 GO:0001568
Blood vessel remodeling	*EPAS1*	GO:0001974
**Differentiation**		
Cell differentiation	*FGF9, RARRES2, FGFR4, FGFR2, EPAS1*	GO:0030154
Epithelial cell differentiation	*TBX1*	GO:0030855
Positive regulation of fat cell differentiation	*RARRES2*	GO:0045600
Osteoblast differentiation	*FGF9, FGFR2, ITGA11*	GO:0001649 GO:0045667
Mesenchymal cell differentiation	*FGFR2*	GO:0048762
**Proliferation**		
Positive regulation of cell proliferation	*TBX1, FGF9, TNFRSF11B, FGFR4, GREM1, NUAK1*	GO:0008284
Positive regulation of mesenchymal cell proliferation	*TBX1, FGFR2*	GO:0002053
Positive regulation of epithelial cell proliferation	*TBX1, FGF9, FGFR2*	GO:0050679
Positive regulation of endothelial cell proliferation	*CCL11*	GO:0001938
**Migration**		
Positive regulation of cell migration	*MMP9, FGFR4, EFS, CCL11*	GO:0030335 GO:0016477
Positive regulation of keratinocyte migration	*MMP9*	GO:0051549
Endothelial cell migration	*GREM1*	GO:0043542
Cell migration involved in sprouting angiogenesis	*GREM1, SLIT2*	GO:0002042
**Others**		
Wound healing	*FGFR2, CELSR1*	GO:0042060
Chondrocyte development	*COL11A1*	GO:0002063

ASC, adipose-derived stem cell; ECM, Extracellular matrix.

## Data Availability

The datasets used and/or analyzed during the current study are available from the corresponding author on reasonable request.

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
