# Peer review of "A Comparative Analysis of the Wound Healing-Related Heterogeneity of Adipose-Derived Stem Cells Donors"

_pharmaceutics, 2022, doi:10.3390/pharmaceutics14102126_

Round 1

Reviewer 1 Report

A necessary and interesting article. Well conducted on a scientific level with a coherent and orderly development. 

Some aspects need to be clarified:

M&M. You start from 3 liposuctions but according to the very accurate comments in the introduction, where is the adipose tissue extracted from in the different donors?, what are the characteristics of the different donors (Sex, Age, Gender, BMI, smoker, alcohol consumption, lifestyle...)?, as the authors know, all these aspects influence the capacities of the cells.

Although it is described that cells are used between passage 1-5, it is known that ASCs have different capacities and markers in that passage range (e.g. CD34) and especially different protein expression (You recognise this point in line 427-428 of the discussion). Another aspect to clarify is the cell ranges per cm2 in the CFU and Proliferation assays, both are very disparate, please clarify. All these dispersions, if they are not homogeneous, could disperse the results, especially gene expression profile between ASCs.

Fig 3. If we analyse the scratch time 0, we observe a higher number of cells in the central region in the groups with ASCs, and especially in ASC-105 and ASC-106, could this affect the final results, please clarify.

Fig 4. Similar comment to figure 3. In addition, figure 4D is not clearly visible and it is difficult to maintain the assertion made in the text.

The discussion is very interesting, although the genes involved in the different steps of wound remodelation and repair are variable according to the literature. You have finally selected a panel of 11 genes, most of them involved in the different steps, but it is not very clear why you select that panel and not an alternative one among the 148 up or downregulation genes.

Author Response

Response to reviewer 1 to “Comparative analysis of wound healing-related heterogeneity of adipose-derived stem cells donors”

We gratefully thank the editor and all reviewers for their time spent making their constructive remarks and useful suggestions, which have enabled us to improve and significantly increased the quality of the manuscript. The reviewers' suggested revisions and comments were carefully considered and incorporated. Below the reviewers' comments are replied to point by point, and the revisions of the manuscript are indicated.

A necessary and interesting article. Well conducted on a scientific level with a coherent and orderly development. 

Some aspects need to be clarified:

M&M. You start from 3 liposuctions but according to the very accurate comments in the introduction, where is the adipose tissue extracted from in the different donors?, what are the characteristics of the different donors (Sex, Age, Gender, BMI, smoker, alcohol consumption, lifestyle...)?, as the authors know, all these aspects influence the capacities of the cells.

Reply: We have added the donor information to the text in section “Cell isolation and culture”.

Adipose tissue was anonymously donated by three healthy subjects undergoing cosmetic liposuction surgery at Aleris Private Hospital, Aalborg, Denmark (Gender: male, female, female; Age: 44, 53, 49; Location: abdomen, abdomen, knee; BMI: 25.4, NA, 23.6.

Although it is described that cells are used between passage 1-5, it is known that ASCs have different capacities and markers in that passage range (e.g. CD34) and especially different protein expression (You recognise this point in line 427-428 of the discussion).

Reply: We apologize for this inconvenience and misunderstanding. Actually, we only used the cells between passages 1-2, the cells were expanded once in the bioreactor, cryopreserved, and after thawing, directly seeded for the different experiments or production of CM. The text has been rewritten to better reflect this.

Hereafter, the ASC cultures were named ASC-101, ASC-105, and ASC-106 and used in passages 1-2 for subsequent procedures.

Another aspect to clarify is the cell ranges per cm2 in the CFU and Proliferation assays, both are very disparate, please clarify. All these dispersions, if they are not homogeneous, could disperse the results, especially gene expression profile between ASCs.

Reply:  The seeding density between the two types of assays varies due to the very nature of the assays. For the proliferation assay, a seeding density was chosen which would allow for cells to proliferate and which enabled us to cover the exponential phase of the proliferating cell cultures before these reached confluency. For the CFU assay, the cells were seeded in a limiting dilution assay to accurately calculate the number of CFUs in the population. This has been clarified in the methods text.

To evaluate the frequency of colony-forming unit (CFU) in the three ASC cultures, each culture was seeded in a limiting dilution assay at densities of 1-30 cells per well in a 96-well plate (Costar, Corning Life Sciences, Tewksbury, MA) and cultured for 14 days with medium change twice a week.

To quantify the proliferation of ASCs, these were seeded with 600 cells/cm2 in a 96-well plate to enable the coverage of the exponential phase of the proliferating cell cultures and cultured in ASC culture medium for 13 days.

Fig 3. If we analyse the scratch time 0, we observe a higher number of cells in the central region in the groups with ASCs, and especially in ASC-105 and ASC-106, could this affect the final results, please clarify.

Reply: The dots visible at the center of the scratches are not cells but artifacts caused by leftover cell membranes. Of the original phase-contrast pictures, this is evident. When analyzing the migration of the cultures, the scratch contour was drawn at each time point, and the difference in total area was calculated. Thereby, the leftover cell membranes did not affect the final results.

Fig 4. Similar comment to figure 3. In addition, figure 4D is not clearly visible and it is difficult to maintain the assertion made in the text.

Reply: We apologize for this inconvenience. We agree that it is difficult to visually assess the tube formation. To avoid bias, we therefore evaluated the branch length the total branching length, extreme, node, and junction number using automated image analysis and presented this data in panel E. We believe the quantitative analysis of the tube formation can give us more information than the visual assessment of the image.

The discussion is very interesting, although the genes involved in the different steps of wound remodelation and repair are variable according to the literature. You have finally selected a panel of 11 genes, most of them involved in the different steps, but it is not very clear why you select that panel and not an alternative one among the 148 up or downregulation genes.

Reply:Thank you for putting this issue forward. We realize that the selection process is not clearly described. Therefore, we have added a new figure (Figure 6) and a supplemental table (S1) giving an overview of the selection process, and extended the description in the methods, results, and discussion. Hopefully, what we did and why is now evident to the reader.

Results section : Selection of putative markers predicting wound healing capacity

To select the most promising markers predicting wound healing capacity, putative predictors were selected based on the analyses above (Fig. 6). In summary, from the pool of 230487 mRNA transcripts (24237 genes), 158 genes (table S1) upregulated in both ASC-105 and -106 compared to ASC-101 were identified (Fig. 5A). Next, a combination of the two different analytical approaches were used to select genes of interest among the upregulated genes. From the first approach based on the gene ontology (GO) analysis of all genes (Fig. 5B), 31 genes associated with wound-healing-related processes were selected. From the second approach based on the heat map analysis, we identified the 14 most significantly upregulated genes (Fig. 5C). Interestingly, between these two approaches, there was an overlap of 10 genes, leading to a total of 35 unique genes (Table 2). Based on a thorough literature review of the 35 genes, we further narrowed the list down to 10 genes of special interest comprising transcription factors (TBX1, EPAS1), membrane proteins (FGFR2, ITGB8, GREM1), and secreted proteins (MMP1, MMP9, COL4A4, FGF9, and CCL11).

Reviewer 2 Report

The authors aim to identify a set of markers to use in the screening of ASCs to treat chronic wounds. Their results have shown that specific regulatory genes associated with ECM and angiogenesis could be used as markers of a superior ASC donor.

The manuscript is clear, easy to understand, and offering potentially important information. However, the following concerns should be addressed before the manuscript can be considered for publication.

Introduction:

-  Line 32 – Please, explain what are the current treatment options.

- Line 46-48 – The authors claim that several production steps could influence the characteristics of the final product. Please, explain better which production steps can affect the final product.

- Line 49 – Please, provide some examples of which production parameters influence the wound-healing properties.

-      Line 57-59 – The authors give some examples from the literature in which ASCs isolated from aged donors have shown reduced potential. Please, include some examples of  the influence of the sex of the donor on the cells’ potential.

-  The authors should discuss more about the importance of potency assays.

Materials and Methods:

-   Line 80 – The authors state that adipose tissue was anonymously donated. Do they know the age and sex of the three donors?

- Line 93 – The authors need to explain better the expansion of cells in the bioreactor.

- Line 116 - Regarding CM, the authors state that CM was prepared with fresh culture medium. Does this medium have FCS?

-  When the CM is collected, is it possible to know the amount of cells that produced this CM? It is important to normalize the CM to the number of cells.

-   Please, replace “cells per cm2” to “cells/cm2”.

-    Line 161 – Please, include what is the “standard culture medium”.

-   Regarding the functional tests, if the CM has FCS, why did the authors use MEM as control and not DMEM with FCS? The control DMEM+FCS should be included in these experiments.

Results & Discussion:

-  The authors should present a picture of the cells from the different donors after 13 days of proliferation.

-  Figure 2D - This graph should have error bars.

-  Line 239-242 - Since the different donors have different proliferative capacity, the  CM collected for further experiments should be normalized to the same number of cells. Otherwise, the results obtained might be affected by the fact that more cells produce more factors that will have a benefit effect.

- Line 291-293 – “Based on the characteristic and functional tests of the ASC cultures derived from distinct donors, it was found that ASC-106 and ASC-105 were consistently superior to ASC- 101”.  This conclusion might be wrong, ASC-106 and ASC-105 might be presenting better properties since these donors were more proliferative so their CM might be enriched with more soluble factors.

-   Please, avoid the repetitive use of the word “Furthermore”.

Author Response

Response to reviewer 2 to “Comparative analysis of wound healing-related heterogeneity of adipose-derived stem cells donors”

We gratefully thank the editor and all reviewers for their time spent making their constructive remarks and useful suggestions, which have enabled us to improve and significantly increased the quality of the manuscript. The reviewers' suggested revisions and comments were carefully considered and incorporated. Below the reviewers' comments are replied to point by point, and the revisions of the manuscript are indicated.

The authors aim to identify a set of markers to use in the screening of ASCs to treat chronic wounds. Their results have shown that specific regulatory genes associated with ECM and angiogenesis could be used as markers of a superior ASC donor.

The manuscript is clear, easy to understand, and offering potentially important information. However, the following concerns should be addressed before the manuscript can be considered for publication.

Introduction:

-  Line 32 – Please, explain what are the current treatment options.

Reply: We have added the following text to our manuscript.

The commonly used clinical treatment for chronic wounds includes inflammation control, wound dressing changing, operative debridement, and flap repair.

- Line 46-48 – The authors claim that several production steps could influence the characteristics of the final product. Please, explain better which production steps can affect the final product.

Reply: Several critical steps in the harvest, isolation, and expansion of ASCs can affect the final ASC-based medicinal products for wound healing. We have now rephrased the sentence and cited our published paper, in which this issue has been thoroughly discussed.

Large-scale production and subsequent adoption of ASC-based medicinal products require attention to better product standardization, and several production steps, from the isolation of starting material to the storage of the product, could influence the characteristics of the final product and the clinical effectiveness [18].

Reference 18: Riis, S.; Zachar, V.; Boucher, S.; Vemuri, M.C.; Pennisi, C.P.; Fink, T. Critical steps in the isolation and expansion of adipose-derived stem cells for translational therapy. Expert Rev. Mol. Med. 2015, 17.

- Line 49 – Please, provide some examples of which production parameters influence the wound-healing properties.

Reply: The use of trypsin or hypoxic treated ASCs have been proven to affect the wound-healing properties of ASC. We have updated our manuscript as per your suggestion.

Indeed, production parameters, such as trypsin and hypoxic treatment, have been shown to influence several of the wound-healing properties, including enhanced immunosuppression [19], secretion of pro-angiogenesis growth factors [20–25], increased endothelial cell growth [20] and angiogenesis [22–25], re-epithelialization [26,27], and extracellular matrix (ECM) production [28].

-      Line 57-59 – The authors give some examples from the literature in which ASCs isolated from aged donors have shown reduced potential. Please, include some examples of  the influence of the sex of the donor on the cells’ potential.

Reply: We have now added more examples to show the role of gender on the differentiation properties of ASCs.

-  The authors should discuss more about the importance of potency assays.

 Reply: We have added a new paragraph heron to the discussion.

Ultimately, the goal would be to identify markers predictive of the wound healing ca-pacity of a stem cell preparation in a clinical setting, such that those markers could be used as potency assays. The importance of developing validated potency assays is un-derscored by the fact that both the European Medicines Agency (EMA) and the U.S Food and Drug Administration require such assays as part of the release criteria after transitioning from early phase trials to an approved medicinal product.

Materials and Methods:

-   Line 80 – The authors state that adipose tissue was anonymously donated. Do they know the age and sex of the three donors?

Reply:  Sorry for the inconvenience, we have now added the donor information to the text.

Adipose tissue was anonymously donated by three healthy subjects undergoing cosmetic liposuction surgery at Aleris Private Hospital, Aalborg, Denmark (Gender: male, female, female; Age: 44, 53, 49; Location: abdomen, abdomen, knee; BMI: 25.4, NA, 23.6.

- Line 93 – The authors need to explain better the expansion of cells in the bioreactor.

Reply: We have added more information about bioreactors and cited a reference in our manuscript, which has described the exact expansion procedure we use very thoroughly.

The obtained cell pellet was handled and expanded in an automated bioreactor Cell Expansion system (Terumo BCT, Lakewood, CO, USA) and frozen for later use as described by Haack-Sørensen et al [40]. For the subsequent analysis of the ASCs, the cells were thawed and cultured in T175 culture flasks (Greiner Bio-one, Frickenhausen, Germany) in alpha-Minimum Essential Medium with GlutaMAX supplemented with 10% fetal calf serum (FCS) and 1% antibiotics (all from Invitrogen), and the TrypLE (Gibco) was used to detach the cells when they reached 70%–80% confluency. Hereafter, the ASC cultures were named ASC101, ASC105, and ASC106 and used in passage 1-2 for subsequent procedures. For an overview of the experimental setup, see figure 1.

- Line 116 - Regarding CM, the authors state that CM was prepared with fresh culture medium. Does this medium have FCS?

Reply: Yes, the ASC culture medium mentioned in the manuscript include alpha-Minimum Essential Medium with GlutaMAX supplemented with 10% fetal calf serum (FCS) and 1% antibiotics.

-  When the CM is collected, is it possible to know the amount of cells that produced this CM? It is important to normalize the CM to the number of cells.

Reply: We seeded the cells at the same density for each group, and as it will take 24 hours for ASCs to properly attach, the cell number in each group should be consistent.

-   Please, replace “cells per cm2” to “cells/cm2”.

Reply: We have replaced all the unit formats in our manuscript.

-    Line 161 – Please, include what is the “standard culture medium”.

Reply: All the ASC culture medium or standard culture medium mentioned in the manuscript include alpha-Minimum Essential Medium with GlutaMAX supplemented with 10% fetal calf serum (FCS) and antibiotics. We have removed the phrase, now only calling it ASC culture medium.

-   Regarding the functional tests, if the CM has FCS, why did the authors use MEM as control and not DMEM with FCS? The control DMEM+FCS should be included in these experiments.

 Reply: The control medium includes alpha-Minimum Essential Medium with GlutaMAX supplemented with 10% fetal calf serum (FCS) and antibiotics. We have now clearly defined this in the text to avoid confusion of the reader.

Results & Discussion:

-  The authors should present a picture of the cells from the different donors after 13 days of proliferation.

Reply: We did the proliferation assay in a 96-well plate and cultured it for 13 days. The below picture shows the ASC proliferated in a 96-well plate for 13 days. It is hard to take a visible and high-quality picture in a 96-well plate for publication, and there is no significant difference from the picture since all the cells have been over confluency after 13 days of culture.

-  Figure 2D - This graph should have error bars.

Reply: In our flow experiment, we only tested each donor for each marker once, so we cannot add error bars in this graph.

-  Line 239-242 - Since the different donors have different proliferative capacity, the  CM collected for further experiments should be normalized to the same number of cells. Otherwise, the results obtained might be affected by the fact that more cells produce more factors that will have a benefit effect.

Reply: We thank the reviewer for this comment, but as we have mentioned above, we seeded the cells at the same high density for each group, and It will take 24 hours for ASCs to attach, so the CM to the number of cells in each group is consistent. As our ASC proliferation assay shows, the proliferation rate of each group is similar in the first three days. Therefore, the results obtained will not be affected by the fact that more cells produce more factors that will have a beneficial effect.

- Line 291-293 – “Based on the characteristic and functional tests of the ASC cultures derived from distinct donors, it was found that ASC-106 and ASC-105 were consistently superior to ASC- 101”.  This conclusion might be wrong, ASC-106 and ASC-105 might be presenting better properties since these donors were more proliferative so their CM might be enriched with more soluble factors.

Reply: We appreciate the reviewer for the constructive comments. We have updated the text based on the suggestion.

Based on the characteristic and functional tests of the ASC cultures derived from distinct donors, it was found that ASC-105 and –106 might present beneficial properties in comparison to ASC-101, which might be explained by their proliferative capacity or other functional attributes. To further understand the underlying mechanism of action behind this, samples from each culture were analyzed by total RNA sequencing to explore the transcriptomic differences between these.

-   Please, avoid the repetitive use of the word “Furthermore”.

Reply: We apologize for this and have revised the text.

Round 2

Reviewer 2 Report

The authors have addressed most of the reviewer's comments.